# Folded Hamiltonian Monte Carlo for Bayesian Generative Adversarial Networks

## Abstract

Generative Adversarial Networks (GANs) can learn complex distributions over images, audio, and data that are difficult to model. We deploy a Bayesian formulation for unsupervised and semi-supervised GAN learning. We propose Folded Hamiltonian Monte Carlo (F-HMC) within this framework to marginalise the weights of the generators and discriminators. The resulting approach improves the performance by having suitable entropy in generated candidates for generator and discriminators' weights. Our proposed model efficiently approximates the high dimensional data due to its parallel composition, increases the accuracy of generated samples and generates interpretable and diverse candidate samples. We have presented the analytical formulation as well as the mathematical proof of the F-HMC. The performance of our model in terms of autocorrelation of generated samples on converging to a high dimensional multi-modal dataset exhibits the effectiveness of the proposed solution. Experimental results on high-dimensional synthetic multi-modal data and natural image benchmarks, including CIFAR-10, SVHN and ImageNet, show that F-HMC outperforms the state-of-the-art methods in terms of test error rates, runtimes per epoch, inception score and Frechet Inception Distance scores.

## 1 Introduction

Generative adversarial networks (GANs) [Goodfellow et al. (2014)] have received traction in the field of deep generative models. The development of GANs covers a wide range from multi-layer perceptrons to the BigGAN framework [Brock et al. (2019)] with residual blocks and self-attention layers (Zhang et al. (2019)) to synthesise realistic images. Despite GAN's effectiveness in generating realistic images, it experiences mode collapse, which occurs when the generator over-optimises for a particular discriminator and the discriminator never learns how to escape the trap. Recent work has focused on alternative metrics such as f-diversities [Nowozin et al. (2016)] or Wasserstein divergences [Arjovsky et al. (2017)] to substitute the Jensen-Shannon divergence inherent in traditional GAN training to alleviating several practical issues.

Saatci & Wilson (2017) recently proposed Bayesian GAN, a probabilistic framework for GANs based on Bayesian inference. It demonstrates how modelling the distribution of generators alleviates mode collapse and motivates the interpretability of learned generators. GAN training measures the full posterior distribution across network weights in a single-mode based on mini-max optimisation. Even if the generator does not recall training instances, samples from the generator are expected to be excessively compact compared with data distribution samples. In fact, as demonstrated by Bayesian GAN, a posterior distribution over the generators' parameters can be vast and highly multi-modal to model the real data distribution by fully reflecting the posterior distribution over the generator and discriminator parameters. In addition, He et al. (2019) proposed the probGAN, which is similar to the Bayesian GAN, iteratively learns a distribution over generators but with a carefully crafted prior. Learning is triggered by a tailored Stochastic Gradient Hamiltonian Monte Carlo (SGHMC) to perform Bayesian inference.

In both Bayesian GAN and ProbGAN, SGHMC is deployed to marginalised the parameters of the generator and discriminators. The foundation of our idea in this paper is to utilise different samples for generator parameters in order to mitigate GAN collapse mode and more efficiently produce data samples with an adequate degree of entropy, especially with high-dimensional and highly correlated

data. To do so, we propose a Folded Hamiltonian Monte Carlo (F-HMC) to replace the SGHMC part of the Bayesian framework. This proposal shares several desired properties with SGHMC, such as 1) being experimentally well-adjusted in training the GAN due to its Hamiltonian dynamics and 2) directly importing parameters such as the learning rate from gradient descent into the sampler. Furthermore, it benefits from the following advantages:

1. F-HMC explores more accurately the target density, especially in the scenario with high-dimensional and highly correlated data.
2. F-HMC converges faster to the target density in terms of lag number.
3. F-HMC provides the practical advantage to the Bayesian GAN method by exploring a rich multi-modal distribution over the weight parameters of generators at an acceptable entropy level.
4. More importantly, because of the parallel composition, F-HMC has an efficient run time in high-dimensional data approximation.

Our main contributions are listed below. We will return to them in the experiment section to highlight each one of them.

1. We propose F-HMC to sample parameters of generators to create candidates from the multi-modal high-dimensional distribution.
2. We have mathematically verified the functionality of F-HMC.
3. Empirical results on a high-dimensional multi-modal synthetic dataset show that the generated samples from our proposed method cover target distribution with more similarity to the target density concerning entropy value.
4. We have shown that using our proposed model, the semi-supervised learning algorithms on natural image datasets (ImageNet, SVHN, and CIFAR10) outperform the state-of-the-art in terms of inception scores (IS) and Frechet Inception Distance scores (FID).

The structure of the paper is as follow: We begin with the problem formulation in Section 2 and enhance the framework by introducing the proposed F-HMC in Section 4. We also have provided the theoretical analysis of F-HMC to generate samples from the desired target. Then, we have demonstrated the performance of our proposed model in Section 5. Finally, Section 6 concludes the paper and discusses future work.

## 2    PROBLEM FORMULATION

Suppose having observed data $D = \{x'_i, y'_i\}_{i=1}^{N}$ from an unknown probability distribution $p_{data}$ where $x'$ describes the input and $y'$ denotes the corresponding label. We would like to estimate $p_{data}$ which is a high-dimensional multi-modal distribution. Bayesian GAN [Saatci & Wilson (2017)] investigates the distributions over the weight parameters of the generators and creates distributions over an infinity space of generators and discriminators, corresponding to every conceivable setting of these weight vectors. We build upon the problem formulation in Bayesian GAN [Saatci & Wilson (2017)] and formulate the posterior as $p(y'|f(x', \alpha))$ where $f(x', \alpha) = Gen(z, \hat{\alpha}_g)$. Here $z$ represents white noise derived from $p(z)$, and $\hat{\alpha}_g$ represents distribution over generator parameters. We denote that parameter set $\alpha$ consisting of two sub parameter set $\hat{\alpha}_g$ related to the generator and $\hat{\alpha}_d$ associated with the discriminators.

We require the weight candidates of generators and discriminators to create candidates for posterior estimation. In this regard, we need to estimate posterior over $\hat{\alpha}_g$, $\hat{\alpha}_d$. Since generators and discriminators are performing min-max optimisation, the parameter over $\hat{\alpha}_g$ and $\hat{\alpha}_d$ are interdependent. First, generator weights $\hat{\alpha}_g$ are sampled from a prior $p(\hat{\alpha}_g|\beta_g)$, and a particular generative neural network is constructed conditioning on these samples. Then, white noise $z$ derived from $p(z)$ is transformed through the network $Gen(z; \hat{\alpha}_g)$ to generate candidate data samples. Conversely, discriminator conditioned on its weights $Disc(.; \hat{\alpha}_d)$ produces the probability that these candidate samples are generated from the data distribution. This process can be stated as follow considering $L$ as the likelihood:

$$p(\hat{\alpha}_g|z, \hat{\alpha}_d) \propto exp\{L(Disc(Gen(z, \hat{\alpha}_g), \hat{\alpha}_d))\}p(\hat{\alpha}_g|\beta_g) \qquad (1)$$

From the discriminator side, we need to form classification likelihood that classifies actual data from the generated samples and can be formulated as:

$$p(\hat{\alpha}_d|z, X, \hat{\alpha}_g) \propto exp\{L(X, \hat{\alpha}_d)\} \times exp\{L(1 - Disc(Gen(z, \hat{\alpha}_g), \hat{\alpha}_d))\}p(\hat{\alpha}_d|\beta_d) \tag{2}$$

Here $p(\hat{\alpha}_d|\beta_d)$ indicates prior for $\hat{\alpha}_d$. By marginalising $z$ from Equations 1 and 2, the equations get updated to $p(\hat{\alpha}_g|\hat{\alpha}_d) = \int p(\hat{\alpha}_g, z|\hat{\alpha}_d)dz$ and $p(\hat{\alpha}_d|\hat{\alpha}_g) = \int p(\hat{\alpha}_d|z, X, \hat{\alpha}_d)dz$. We can approximate the posterior over $\hat{\alpha}_g$ and $\hat{\alpha}_g$ by iteratively sampling from $p(\hat{\alpha}_g|\hat{\alpha}_d)$ and $p(\hat{\alpha}_d|\hat{\alpha}_g)$. Therefore we can have the corresponding generators and discriminators to generate candidate samples from the multi-modal high dimensional distribution ($p_{data}$). This paper proposes F-HMC as an efficient sampling strategy, especially when the target is high-dimensional and highly correlated.

## 3 BACKGROUND

This section briefly reviews data sampling basics, such as the Hamiltonian Monte Carlo sampler and Stochastic Hamiltonian Monte Carlo derived from [Chen et al. (2014)]

Suppose one wants to sample from the posterior distribution of $X$ given a set of independent observations $x \in D$:

$$p(X|D) \propto \exp\left(-U(X)\right) \tag{3}$$

where the potential energy function U is given by

$$U = -\sum_{x \in D} \log p(x|X) - \log p(X) \tag{4}$$

Hamiltonian Monte Carlo (HMC) is a method for efficiently exploring the state space by proposing samples of $X$ in a Metropolis-Hastings (MH) framework. By incorporating auxiliary momentum variables, $V$, these suggestions are generated from a Hamiltonian system. HMC creates samples from a joint distribution of $(X, V)$ to sample from $p(X|D)$:

$$\pi(X, V) \propto \left(\exp(-U(X) - \frac{1}{2}V^T M^{-1} V\right) \tag{5}$$

The samples of $X$ have a marginal distribution $p(X|D)$ if the resultant samples of $V$ are simply discarded. Here $M$ is a mass matrix that, coupled with $V$, defines a kinetic energy term. $H(X, V) = U(X) - \frac{1}{2}V^T M^{-1} V$ defines the Hamiltonian function. H intuitively calculates the total energy of a physical system by utilising position variables $X$ and momentum variables $V$. To propose samples, HMC simulates Hamiltonian dynamics.

$$\begin{cases} dX = M^{-1}V dt \\ dV = -\nabla U(X) dt \end{cases} \tag{6}$$

Furthermore, SGHMC is based on the idea of combining stochastic optimisation with a first-order Langevin dynamic MCMC technique, demonstrating that adding the "right amount" of noise to stochastic gradient ascent iterates results in samples from the target posterior as the step size is annealed. SGHMC accomplishes this by including a "friction" term in the momentum update:

$$\begin{cases} dX = M^{-1}V dt \\ dV = -\nabla U(X) dt - BM^{-1}V dt + \mathcal{N}(0, 2Bdt) \end{cases} \tag{7}$$

## 4 F-HMC MODEL

This section proposes F-HMC as an efficient and scalable sampling strategy, particularly when the target is multi-modal and highly correlated (Contribution 1). Additionally, we have presented the mathematical analysis of F-HMC (Contribution 2) in section 4.2 to prove that the F-HMC samples from the equivalent target distribution specified in Equation 7.

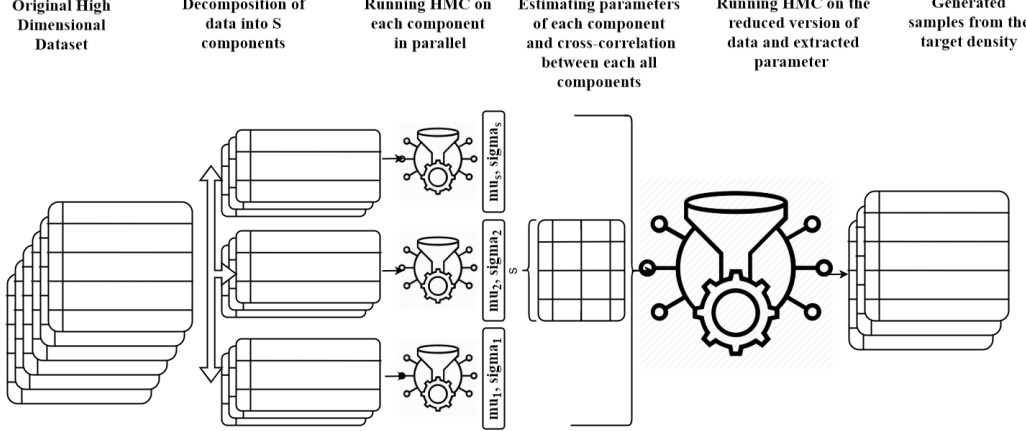

Figure 1: The pipeline of the F-HMC model which is consists of 4 stages. The first stage is decomposed into S regions using a Gaussian mixture. The second stage is running the regions in parallel and estimating their corresponding density parameter. The third stage is running another HMC on data coming from the last stage with respect to the cross-correlation between components to ensure reliable flow between different regions and finally generating data from the target distribution.

## 4.1 METHODOLOGY

In the scenario where the gradient elements are on dramatically different scales and highly correlated, the SGHMC cannot efficiently explore the target density. As a pragmatic approach, we propose F-HMC, a parallelised algorithm that, instead of finding a single chain that samples from the whole distribution, combines samples from several chains. Chains are running in parallel, each exploring a different region of the state space (e.g., a few modes only). F-HMC assumes that a Gaussian mixture approximates the target density $pi$. In that case, each mixture component represents a reliable proposal in a specific region of the sample space $S \subset R^d$. Assume $pi$ has $J$ modes and consider its approximation via the mixture model as follows:

$$q(x) = \sum_{j=1}^{J} \mathcal{N}_d(x; \mu^j, \sigma^j) \tag{8}$$

where $\mathcal{N}_d(x; \mu^j, \sigma^j)$ is the probability density of a d-variate Gaussian distribution with $\mu$ mean and $\sigma$ covariance matrix. We define the partition $\hat{S} = \cup_{k=1}^{J} S^j$ using the mixture representation in 8, so that $pi$ is more similar to $\mathcal{N}_d(x; \mu^j, \sigma^j)$ on each set $S^{(j)}$ than any other distribution entering 8, which is formulated as follow:

$$S_n^{(j)} = \{x : argmax\mathcal{N}(x; \mu_n^{(j')}, \sigma_n^{(j')}) = j\} \tag{9}$$

The approximation 8 along with the regions defined in 9 enable us to determine the proposal distribution of F-HMC. One may consider proposal distribution as:

$$Q(x) = \sum_{j=1}^{J} S_n^{(j)} \mathcal{N}(x, \mu_n^{(j)}, \sigma_n^{(j)}) \tag{10}$$

In other words, we would utilise the dominating component of the mixture as a proposed distribution in each area $S_n^{(j)}$. While such a concept may have acceptable local properties, it may not ensure optimal flow across various regions. Therefore, we are using a second or (fold) HMC on top of these samplers concerning the cross-correlation of partitions to allow a flow between different regions. Thus the F-HMC proposal distribution models are as follow:

$$Q'(x) = \sum_{j=1}^{J} Q(x)\mathcal{N}(x, \hat{\mu}^{(j)}, \hat{\sigma}^{(j)}) \tag{11}$$

where $\hat{\sigma}$ is calculated with respect to cross-correlation between all components. Figure 1 demonstrates the functionality of the F-HMC; since the first part is running on parallel and the second fold is running on a reduced number of dimensions, the overall execution time, especially on high dimensional setup, improves. Furthermore, because two HMC samplers thoroughly examine the data, the system's entropy will be satisfactory. Algorithm 1 shows one iteration of the Bayesian learning for the generator parameters using our proposed model. Here $\theta$ is the friction term for HMC, and $\eta$ is the learning rate. K shows the number of iterations in Bayesian GAN, and $I$ shows the number of F-HMC iterations, and S indicates the number of components in the F-HMC.

---

**Algorithm 1** One iteration of the generator in Bayesian GAN set up with our proposed F-HMC

---

1: **Input:** $\{\hat{\alpha}_g^{i,k}\}$ and $\{\hat{\alpha}_d^{i,k}\}$ from preceding iteration and number of $\theta$ as HMC friction term, $\eta$ for the learning rate, and $I$ as the number of F-HMC iterations.
2: **for each** $i \leftarrow 1$ to $I$ **do**
3:      $z \sim p(z)$                                        ▷ sampling white noise z from its prior
4:      **for each** $s \leftarrow 1$ to $S$ **do**                         ▷ running the decomposition
5:          $\mu_s \leftarrow HMC(logp(\hat{\alpha}_g^{i,k,s})|z, \hat{\alpha}_d^s)$
6:          append $\mu_s$ to the $\hat{\mu}$
7:      **end for each**
8:      $\hat{\sigma} \leftarrow$ calculate covariance between S components
9:      $q \leftarrow HMC(\hat{\mu}, \hat{\sigma})$            ▷ merging all back to sample from the target density
10: **end for each**
11: $n \sim \mathcal{N}(0, 2\theta\eta I)$
12: $v \leftarrow (1-\theta)v + \eta q + n$
13: $\hat{\alpha}_g^{i,k} \leftarrow \hat{\alpha}_g^{i,k} + v$                            ▷ updating $p(\hat{\alpha}_g)$ sample set
14: **Output:** generated samples for $\hat{\alpha}_g$

---

## 4.2 THEORETICAL ANALYSIS

The purpose of this section is to demonstrate that the probability density of data created using F-HMC is similar to the target distribution of SGHMC. As a result, we can confirm that F-HMC samples are mathematically valid (Contribution 2). Therefore, we can replace the F-HMC as a valid sampler with SGHMC in the Bayesian GAN design. In the Experiment section, we illustrated how this replacement improved the outcomes. Considering earlier setup outlined in Background section 3,$V$ in Equation 7 , converges to a stationary distribution [Chen et al. (2014)] given by the following formula:

$$V \sim \mathcal{N}(\underbrace{MB^{-1}\nabla U(X)}_{\hat{\mu}}, \overbrace{M}^{\hat{\sigma}})$$ (12)

By taking the F-HMC methodology into account, the $V$ in Equation 7 can be decomposed as the following matrix:

$$\begin{bmatrix} V_1 \\ V_2 \\ . \\ . \\ V_S \end{bmatrix} \sim \mathcal{N}\left(\begin{bmatrix} \hat{\mu}_1 \\ \hat{\mu}_2 \\ . \\ . \\ \hat{\mu}_S \end{bmatrix}, \begin{bmatrix} \hat{\sigma}_{11} & . & . & \hat{\sigma}_{1S} \\ . & . & . & . \\ . & . & . & . \\ \hat{\sigma}_{S1} & . & . & \hat{\sigma}_{SS} \end{bmatrix}\right)$$ (13)

**Theorem:** $\pi(X, V) \propto \exp(-H(X, V))$ *is the unique stationary distribution of the dynamics described in Equation 7.*

**Proof**: let $W = \begin{bmatrix} 0 & -I \\ I & 0 \end{bmatrix}$ and $R = \begin{bmatrix} 0 & 0 \\ 0 & B \end{bmatrix}$, Equation 7 along with the proposed decomposition in Equation 13 can be written in the following form:

$$\begin{aligned} d\begin{bmatrix} X \\ V \end{bmatrix} &= d\begin{bmatrix} X_1 & X_2 & . & . & X_S \\ V_1 & V_2 & . & . & V_S \end{bmatrix} \\ &= -\begin{bmatrix} 0 & -I \\ I & B \end{bmatrix}\begin{bmatrix} \nabla U(X_1) & \nabla U(X_2) & . & . & \nabla U(X_S) \\ M^{-1}V_1 & M^{-1}V_2 & . & . & M^{-1}V_S \end{bmatrix} + \mathcal{N}(0, 2Rdt) \\ &= -[W+R]\nabla H(X,V)dt + \mathcal{N}(0, 2Rdt) \end{aligned}$$ (14)

We employ the Fokker-Planck Equation (FPE) (Risken (1992)) to describe the temporal evolution of the probability density function in relation with a Stochastic Differential Equation (SDE) that defines the development of the distribution on the random variable under specified stochastic dynamics. Under Hamiltonian dynamics, the random variables in our situation are position variable X and momentum variable V. Consider the following SDE:

$$d\omega = g(\omega)dt + \mathcal{N}(0, 2R(\omega)dt) \tag{15}$$

$\rho_t(\omega)$ is the distribution of $\omega$ governed by Eq. 15. we also Consider $J_i(.)$ as $\partial_i(.)$ by using FPE the equation evolves under the following formula:

$$J(\rho_t(\omega)) = -\sum_{i=1}^{n} J_{\omega_i}(g_i(\omega)\rho_t(\omega)) + \sum_{i=1}^{n}\sum_{j=1}^{n} J_{\omega_i}(J_{\omega_j}(R_{ij}(\omega)\rho_t(\omega))) \tag{16}$$

We can rewrite Equation 16 in the following compact form:

$$J(\rho(\omega)) = -\nabla^T[g(\omega)\rho_t(\omega)] + \nabla^T[R(\omega)\nabla\rho_t(\omega)] \tag{17}$$

where $\nabla^T[g(\omega)\rho_t(\omega)] = \sum_{i=1}^{n} J_{\omega_i}(g_i(\omega)\rho_t(\omega))$, and $\nabla^T[R(\omega)\nabla\rho_t(\omega)] = \sum_{ij} J_{\omega_i}(R_{ij}(\omega)J_{\omega_j}(\rho_t(\omega)) = \sum_{ij} J_{\omega_i}(R_{ij}(\omega)J_{\omega_j}\rho_t(\omega)) + \sum_{ij} J_{\omega_i}(J_{\omega_j}(R_{ij}(\omega))\rho_t(\omega)) = \sum_{ij} J_{\omega_i}(J_{\omega_j}(R_{ij}(\omega)\rho_t(\omega)))$.

From Equation 17 and considering the variable $\omega = (X, V)$ and $g(\omega) = -[R + W]\nabla H(X, V)$ and $R(\omega) = R(X, V) = R = \begin{bmatrix} 0 & 0 \\ 0 & B \end{bmatrix}$, the distribution evolution under dynamic system in Equation 14 can be written as follow:

$$J(\rho_t(X, V)) = \nabla^T\{[R + W][\rho_t(X, V)\nabla H(X, V) + \nabla\rho(X, V)]\} \tag{18}$$

Since $\pi(X, V) \propto \exp(-H(X, V))$, we can verify that the $\pi(X, V)$ is invariant under Equation 18 by calculating $[e^{-H(X,V)}\nabla H(X, V) + \nabla e^{-H(X,V)}] = 0$. Therefore, we can confirm that the dynamics given in Equation 13 have similar invariance properties to that of the original Hamiltonian dynamics of Equation 7.

## 5 EXPERIMENTS

We implemented the proposed model using Pymc3 [1] and report its performance on generating samples from complex distributions in section 5.1. We have examined the model's performance in marginalising the generators' parameters on synthetic and natural image datasets such as SVHN [Netzer et al. (2011)], CIFAR 10 [Krizhevsky (2009)] and ImageNet [Deng et al. (2009)] in section 5.2 and 5.3, respectively. We have compared our results with WDCGAN [Arjovsky et al. (2017)], DCGAN, 10DCGAN (which is a fully supervised convolutional neural network composed of ten DCGANs constructed by ten random subsets with 80% of the size of the training set, [Radford et al. (2016)]), Bayesian GAN [Saatci & Wilson (2017)] and ProbGAN [He et al. (2019)] on supervised and semi-supervised tasks with four different numbers of labelled examples. For a fair comparison, each model has the same number of generators and discriminator with the same architecture.

### 5.1 F-HMC PERFORMANCE

To evaluate the performance of F-HMC and compare it against SGHMC, we have designed a set of experiments. First, we use Normalising Flows [Kobyzev et al. (2020); Rezende & Mohamed (2016)] as a rich family of distributions to examine F-HMC and SGHMC's abilities to explore complex distribution. Figure 2 shows the potential energy of the rich target distribution and the generated candidates using SGHMC and F-HMC. F-HMC successfully covers the target distribution (Advantage 1). Second, we have measured auto-correlation between the samples generated in each sampler as a metric to show the power of the sampler in exploring the target distribution. The more precise the sampler, the faster the auto-correlation reaches zero. Figure 3 shows auto-correlation in F-HMC drops faster to zero than SGHMC in terms of lag number (Advantage 2).

---

[1]https://docs.pymc.io/

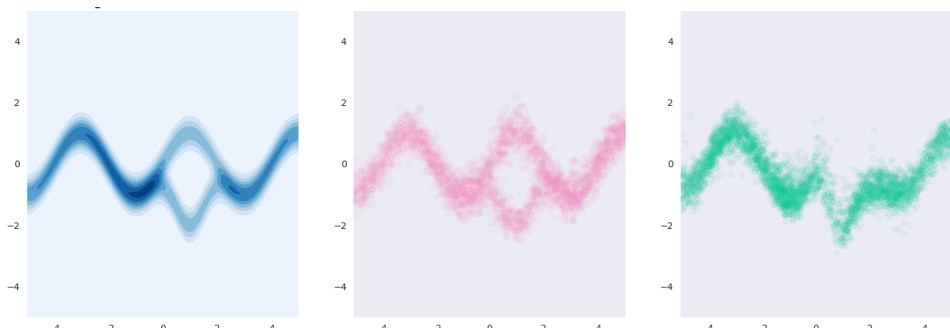

Figure 2: The graph on the left shows the potential distribution of the target. The middle graph shows F-HMC samples exploring the target, and the right graph shows the SGHMC samples. Both samplers converged to the target distribution, but the F-HMC covered the target more accurately than SGHMC.

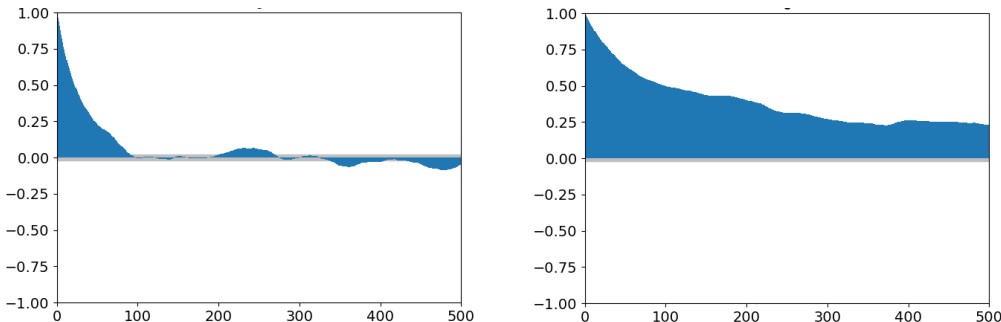

Figure 3: Auto-correlation between samples while the samplers explore the target distribution with 100 dimensions. The left graph shows auto-correlation of samples using the F-HMC sampler while S is 25. The right graph shows the auto-correlation on the same setup using SGHMC. F-HMC converges faster than SGHMC to the target in terms of lag number

## 5.2 HIGH-DIMENSIONAL MULTI MODAL SYNTHETIC DATASET

We present experiments on a multi-modal synthetic dataset to test inferring a multi-modal posterior $p(\hat{\alpha}_g|D)$. This experiment shows F-HMC's ability to explore a set of generators' parameters with proper entropy and different complementary properties to encapsulate a rich data distribution. We fit a regular GAN, Bayesian GAN, and our proposed model to a dataset with D = 100 and 500. The generator for all models is a two-layer neural network: 10-1000-100, fully connected, with ReLU activation. The red samples in Figure 4 depict the target data, whereas the green samples depict the corresponding generated data. The experiments on D = 100 are shown in the first two rows, while the results on D = 500 are shown on two lower rows. The name of generating sampler is displayed on the right side of each row. We can ensure that both strategies (BGAN and F-HMC GAN) cover the intended distribution simply by comparing them visually. Even the visual comparison is insufficient to detect more outstanding performance in D = 100. Still, it is evident in D = 500 that F-HMC provides a more desirable match to the target distribution as the dimension increases (Advantage 3 and Contribution 3). Figure 5 shows the comparison of the performance of GAN, F-HMC GAN, and Bayesian GAN in terms of Jensen-Shannon divergence. The experiment estimates the similarity of the probability distribution of generated data to the original data and the level of entropy and confirms that the F-HMC exceeds other models (Advantage 3 and Contribution 3).

## 5.3 NATURAL IMAGE DATASET

We used a 5-layer network architecture for GAN's generator in all experiments on the natural images' datasets. The corresponding discriminator for supervised GAN is a 5-layer 2-class DCGAN, and we have used a 5-layer, K + 1 class DCGAN for a semi-supervised GAN performing classifica-

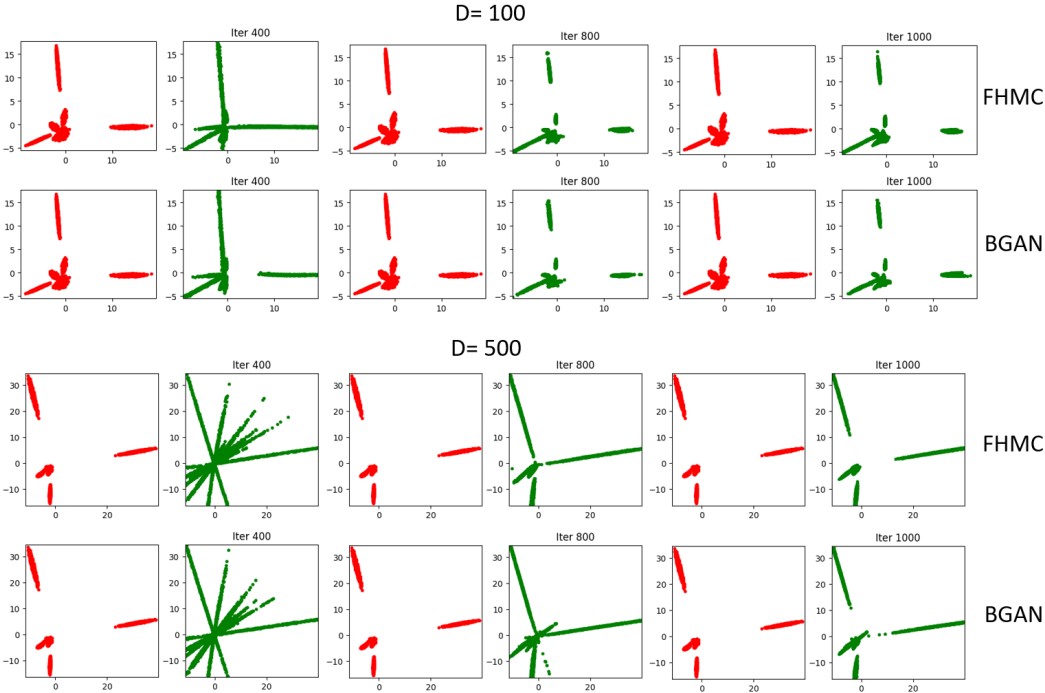

Figure 4: samples drawn from $p_{data}(x)$ and visualised in 2D. the red colour graph shows real data, and the green colour graph shows generated samples. The first two upper rows show the experiment on D = 100, and the two lower rows show D = 500. The name of generating sampler is shown on the right side of each row. The graphs generated by F-HMC have more visual similarity to the actual target, especially when the D = 500.

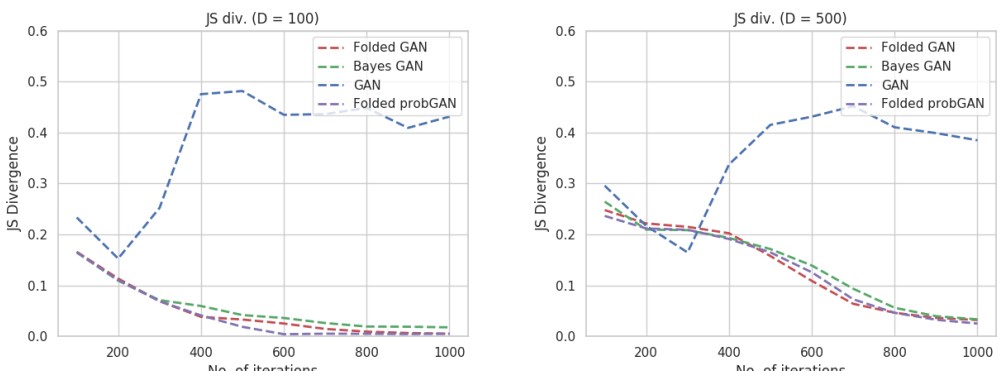

Figure 5: The Jensen-Shannon divergence between $p_{data}(x)$ and the number of iterations of model training. The left graph shows the experiment while D = 100, and the right graph shows the same experiment with D = 500. We can confirm that F-HMC exceeds other models in generating data more similar to the target concerning entropy.

tion over K classes (see Saatci & Wilson (2017) for further details about Bayesian GAN structure)). To evaluate the performance of our proposed model, we have employed experiments in three measurement levels: 1-performance metric in supervised and semi-supervised learning using test error rate. 2-run time per epoch in minutes by running all the models on a single GPU. 3-quality of generated images in terms of IS and FID scores. FID [Heusel et al. (2017)] is a measure that calculates the distance between vectors derived for real and synthetic images. IS [Salimans et al. (2016)] is an objective measure for assessing the quality and diversity of generated images.

Table 1: Supervised and semi-supervised learning results for image benchmarks. the $N_s$ shows number of labelled examples.

| $N_s$ | Supervised | DCGAN10 | W-DCGAN | BayesGAN | probGAN | F-HMC GAN |
|---|---|---|---|---|---|---|
| CIFAR-10 | | | | | | |
| 500 | $65.1 \pm 2.3$ | $30.9 \pm 2.7$ | $55.8 \pm 2.9$ | $30.5 \pm 2.3$ | $30.1 \pm 2.8$ | $\mathbf{30.0 \pm 3.1}$ |
| 1000 | $54.6 \pm 2.1$ | $29.1 \pm 2.4$ | $48.8 \pm 3.2$ | $\mathbf{27.4 \pm 2.1}$ | $27.7 \pm 3.1$ | $27.6 \pm 2.8$ |
| 2000 | $52.4 \pm 2.4$ | $26.8 \pm 3.3$ | $37.9 \pm 2.5$ | $24.2 \pm 1.9$ | $28.3 \pm 2.5$ | $\mathbf{23.9 \pm 2.3}$ |
| 4000 | $48.1 \pm 1.0$ | $24.7 \pm 2.7$ | $28.2 \pm 2.9$ | $22.3 \pm 3.2$ | $21.7 \pm 2.8$ | $\mathbf{20.7 \pm 2.6}$ |
| SVHN | | | | | | |
| 1000 | $55.1 \pm 3.3$ | $30.8 \pm 2.3$ | $30.1 \pm 1.9$ | $28.7 \pm 3.1$ | $\mathbf{26.4 \pm 2.1}$ | $26.6 \pm 2.2$ |
| 2000 | $36.7 \pm 2.63$ | $17.9 \pm 1.7$ | $27.2 \pm 2.6$ | $14.2 \pm 2.8$ | $14.1 \pm 2.6$ | $\mathbf{13.7 \pm 1.8}$ |
| 4000 | $28.2 \pm 3.13$ | $15.8 \pm 1.4$ | $25.1 \pm 2.8$ | $12.7 \pm 2.9$ | $13.5 \pm 1.7$ | $\mathbf{11.7 \pm 1.4}$ |
| 8000 | $21.1 \pm 2.2$ | $15.1 \pm 1.3$ | $20.1 \pm 1.9$ | $9.2 \pm 1.8$ | $11.4 \pm 1.8$ | $\mathbf{8.9 \pm 0.9}$ |
| ImageNet | | | | | | |
| 1000 | $57.6 \pm 4.2$ | $53.4 \pm 3.1$ | $55.7 \pm 3.7$ | $48.9 \pm 4.3$ | $47.8 \pm 4.6$ | $\mathbf{43.8 \pm 4.4}$ |
| 2000 | $42.3 \pm 3.5$ | $38.7 \pm 2.5$ | $40.6 \pm 3.1$ | $34.6 \pm 4.6$ | $34.5 \pm 3.8$ | $\mathbf{33.6 \pm 3.7}$ |
| 4000 | $40.1 \pm 3.6$ | $31.8 \pm 2.1$ | $35.5 \pm 2.9$ | $27.8 \pm 3.8$ | $26.8 \pm 3.2$ | $\mathbf{25.9 \pm 3.5}$ |
| 8000 | $36.8 \pm 4.1$ | $28.3 \pm 1.8$ | $34.3 \pm 2.7$ | $24.4 \pm 3.1$ | $24.1 \pm 2.8$ | $\mathbf{22.7 \pm 2.7}$ |

Table 2: IS (higher is better), FID (lower is better) both trained with WGAN objective and run time (epochs in minutes) results on natural images datasets.

| Dataset | Score | 10DCGAN | BayesGAN | probGAN | F-HMC GAN |
|---|---|---|---|---|---|
| CIFAR10 | IS | 7.78 | 7.69 | 7.72 | **7.79** |
| | FID | 23.81 | 24.75 | 24.63 | **23.73** |
| | Runtime | 143 | **91** | 94 | 93 |
| SVHN | IS | 8.34 | 8.27 | 8.19 | **8.31** |
| | FID | 49.61 | 51.78 | 52.32 | **47.21** |
| | Runtime | 151 | 98 | **89** | 94 |
| ImageNet | IS | 8.41 | 8.51 | 8.56 | **8.59** |
| | FID | 30.2 | 29.78 | 28.12 | **27.83** |
| | Runtime | 671 | 358 | 349 | **336** |

Table 1 demonstrates supervised and semi-supervised learning results for all image benchmarks. Our proposed model mainly outperforms BayesGAN, probGAN, W-DCGAN, and 10-DCGAN in terms of test error rate (Contribution 4). F-HMC shows its substantial impact when running on higher-dimensional data (ImageNet) due to the parallel composition of F-HMC; it can efficiently explore higher dimension data. Table 2 shows the generated images' quality and the run time of the models. The quality of images increases by using F-HMC. It is observed that the run time enhances when running the model on a higher dimension. We perceive that in lower-dimensional data (CIFAR, SVHN), the Bayesian GAN and probGAN's run time is more satisfying. Once we run the models on higher-dimensional data (ImageNet), the run time improves in F-HMC. The parallel composition of F-HMC makes the over epoch run time less than exploring the whole dimensions at once in Bayesian GAN and probGAN (Advantage 4).

## 6 CONCLUSION

Folded Hamiltonian Monte Carlo is presented in this paper as a scalable strategy in sampling high-dimensional, highly correlated data to improve Bayesian GAN in producing synthetic images/generating data by marginalising the weights of the generators and discriminators. We demonstrated that F-HMC converges faster and adapts to higher-dimensional inputs with more significant similarity to the target data. The theoretical and mathematical analysis of F-HMC is presented which confirms its functionality. F-HMC outperforms the state-of-the-art in terms of test error rates, runtimes per epoch, IS, and FID scores when evaluated on synthetic high-dimensional multi-modal data and natural image benchmarks, such as CIFAR-10, SVHN, and ImageNet. Despite the notable improvement that F-HMC can bring in sampling target density, its hyperparameters affect its enforcement. Nested sampling methods [Betancourt et al. (2011)] sample from the likelihood space instead of sample space and are more likely to land on a better set of parameters. Future directions include the adoption of a nested sampling method, e.g. restricted Hamiltonian Monte Carlo methods, to help the improvement of the outcomes.

# 7 REPRODUCIBILITY

The experiments presented in the paper are designed to be reproducible and easy to extend. The notebook provides instructions on how to install the packages as well as the code required to run the experiments and the link to download the necessary datasets.

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

## A    APPENDIX

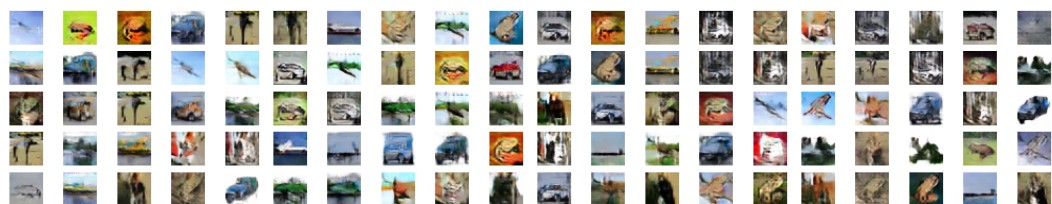

Figure 6: Data samples for the CIFAR10 generated using F-HMC strategy

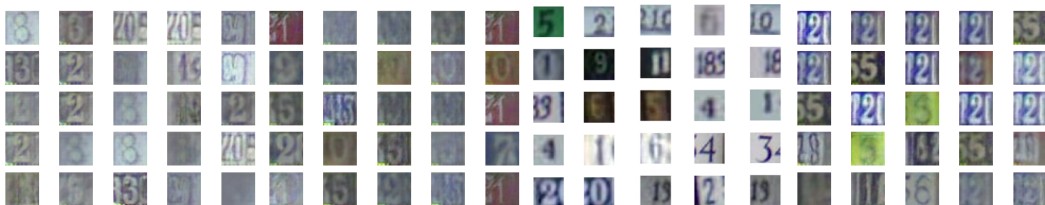

Figure 7: Data samples for the SVHN generated using F-HMC strategy

