# OpenReview forum: "Folded Hamiltonian Monte Carlo for Bayesian Generative Adversarial Networks"
_ICLR.cc/2022/Conference — ICLR 2022 Submitted_

### Official Review · Reviewer_jjmZ · 2021-10-27

**Correctness:** 2
**Technical Novelty And Significance:** 2
**Empirical Novelty And Significance:** 2
**Recommendation:** 3
**Confidence:** 3

**Main Review:**

Overall, the idea of parallelizing HMC is an interesting one. However, I do have some concerns about clarity and the results of the paper

1. Section 4, and particularly 4.1 (which expands on the main contribution of the paper) is confusing and very hard to follow:
- the algorithm uses a "parameter decomposition" to "divide tasks into blocks": how is parameter decomposition carried out? What does it mean to divide updating tasks into blocks? Is this decomposition different for every update of the networks, or is it determined before training begins?
- the components are then "run in parallel" -- Figure 1 implies that this HMC run simultaneously on different subsets of data dimensions, but Algorithm 1 implies that the subsets are for parameters.
- HMC fit a distribution with "Gaussian likelihoods" -- Equation (1) states that the likelihood is proportional to the exponential of the GAN loss function: does the algorithm use a Gaussian approximation for this likelihood?
- Finally, the "$\mu$s and $\sigma$s are given [...] cross-correlation between all components" -- it is very hard to understand what the algorithm does from this sentence.

Overall, from the explanation in section 4.1, it is quite difficult to figure out what exactly the algorithm does, and therefore very hard to judge its merits and demerits.


2. The theoretical analysis in section 4.2 looks at whether HMC performed in parallel on subsets of data dimensions converges to the same distribution as performing HMC on the full data. However, it is very hard to see what the different parts of the proof achieve, or how they are connected to HMC for the Bayesian GAN formulation without some more explanation. For example, it is very hard to follow the logic for deriving Equations 11-14, when all the meat in proving convergence seems to lie in the inline equation at the very end of section 4.2


3. The scores for F-HMC GAN are not better than BayesGAN or probGAN across the board on all applications presented in the paper, and are sometimes only marginally better:  e.g IS scores on CIFAR-10 and ImageNet in Table 2 differ in the 3rd significant digit, the JSD traces for BayesGAN and  FHMC GAN are quite similar and seem to converge to the the same value in Figure 5, etc. Although this does not automatically disqualify the use of FHMC-GAN since it certainly has an advantage in terms of lower runtime, it seems also a bit of an exaggeration to claim the "superiority of FHMC [...] in generating data"


### Minor comments:
- there were numerous typos throughout the manuscript: it would be good to have a thorough edit
- In Eqn 1, LHS ia posterior conditioned on $\alpha_d$ but RHS has a likelihood dependent on $\hat{\alpha}_d$
- The notation in section 3 is confusing: previously $x$ was used to denote target data, and $X$ as the set of all $x$
- The theorem statement in section 4.2 refers to the "dynamics described in Eqn 9" -- should this be Eqn 7?
- In Table 1, the score for probGAN with 4000 samples from ImageNet has the lowest value (25.8) and should be in bold.

**Summary Of The Paper:**

The paper proposes an extension of BayesGAN (Saatci and Wilson, 2017): BayesGAN learns to sample generator and discriminator parameters from a posterior distribution conditioned on target data using Hamiltonian Monte Carlo sampling; this paper proposes using a modified version dubbed folded-Hamiltonian Monte Carlo to speed up the sampling process and make it scalable for high-dimensional data.

**Summary Of The Review:**

It was hard to judge the merits of the algorithm and the contributions of the paper due to a lack of clarity in the explanations, and (it seemed to me) an exaggeration of the results.
I would be happy to re-evaluate if the authors could provide a clearer explanation of the method and results.

---

> ### Author Response · Authors · 2021-11-22
> **Response to AnonReviewer3**
>
> We would like to thank the reviewer for their in-depth comments and concerns, and questions. It helped us a lot to find the unclear part and improve our work. We have addressed the concerns as follow:
>
> 1. Section 4, 4.1 is confusing
>
> We appreciate the reviewer's concern about clarifying the paper's core and explaining the F-HMC in the formula instead of words. To address that, we have updated sections 4.1, 4.2 to present the mathematical formulation of F-HMC, including how we divide the data in S and more details of parameter mu and sigma and how we merge them back. We have listed them here for your convenience.
>
> 2. How to decompose data in S
>
> F-HMC assumes that the target density $pi$ is approximated using a mixture of Gaussians.  In that case, it is possible to affirm that each mixture component represents a reliable proposal in a given region of the sample space $S⊂R^d$.  To clarify the idea, suppose $pi$ has J modes and consider its approximation by the mixture model as follow:
>
> $q(x) = \sum_{j=1}^{J} \mathcal{N}_d (x;\mu^j, \sigma^j)$                 (eq.1)
>
> where
> $\mathcal{N}_d(x;\mu^j,\sigma^j)$
>
>  is the probability density of a d-variate Gaussian distribution with mean μ and covariance matrix σ.  Using the mixture representation (eq.1), we define the partition
>  $\hat{S} = \cup_{k=1}^J S^{j}$   so that $pi$ is more similar to $\mathcal{N}_d (x;\mu^j, \sigma^j)$ on each set  $S^{(j)}$ than any other distribution entering (eq.1) which is formulated as follow:
>
> $S_n^{(j)} = [x: argmax \mathcal{N}(x;\mu_n^{(j')}, \sigma_n^{(j')}) = j]$            (eq.2)
>
> The approximation (eq.1) along with the partition defined in (eq.2) allows us to determine the proposal distribution of F-HMC.
> One may consider proposal distribution as:
>     $Q(x) = \sum_{j=1}^{J} S_n^{(j)} \mathcal{N}(x,\mu_n^{(j)},\sigma_n^{(j)})$
> In other words, we would utilise the dominating component of the mixture as a proposed distribution in each area $S_n^{(j)}$. While such a concept may have acceptable local properties, it may not ensure optimal flow across various regions. Therefore, we are using a second or (fold) HMC on top of these samplers concerning the cross-correlation of partitions to allow a flow between different regions. Thus the F-HMC proposal distribution models are as follow:
>      $Q'(x) = \sum_{j=1}^{J} Q(x) \mathcal{N}(x,\hat{\mu}^{(j)},\hat{\sigma}^{(j)})$
> where $\hat{\sigma}$ is calculated with respect to cross-correlation between all components.
>
>
> 3. Figure 1 implied different subset of data while the algorithm says parameters
>
> Thank you for your feedback. Figure one implies parallel samplers running on different regions of data. We have updated the caption for Figure 1 to avoid confusion and be more precise.
>
> 4. HMC using a gaussian approximation for the likelihood? And mu and sigma and details of core body
>
> F-HMC assumes that the target density $pi$ is approximated using a mixture of Gaussians.  In that case, it is possible to affirm that each mixture component represents a reliable proposal in a given region of the sample space $S⊂R^d$.  To clarify the idea, suppose $pi$ has J modes and consider its approximation by the mixture model as follow:
>
> $q(x) = \sum_{j=1}^{J} \mathcal{N}_d (x;\mu^j, \sigma^j)$                 (eq.1)
>
> where
> $\mathcal{N}_d(x;\mu^j,\sigma^j)$
>
>  is the probability density of a d-variate Gaussian distribution with mean μ and covariance matrix σ.
>
> 5. How section 4.2 is connected to Bayesian GAN, what is it achieving
>
> In section 4.2 we tried to prove that F-HMC as a sampler converges to the target distribution. We wanted to determine that F-HMC's sampling design is mathematically sound. The connection with Bayesian GAN is that we suggest F-HMC as a mathematically reliable sampler instead of SGHMC in Bayesian GAN since SGHMC struggles in high dimensions with high correlations scenarios.
>
> 6. Scores are marginally better, which does not disqualify the design, but it is exaggerating to say the design is superior
>
> Thanks for your comment and we have updated the wording in the manuscript.
>
> 7. Minor comments:
>
> We appreciate the reviewer for their concerns explained in the Minor comment section. We have updated the related parts in the revised manuscript.

---

> > ### Comment · Reviewer_jjmZ · 2021-11-29
> > **Thank you**
> >
> > I thank the authors for their responses and for the updated manuscript.
> >
> > Section 4.1 is clearer now: however, there are still gaps prevent understanding how the method works.
> >  For example:
> > - Is $pi$ the same density as $\pi(X, V)$ from section 3?
> > - The definition of the partitions $S$ seems tautological: the mixture of Gaussians are defined around J modes, and which are used to define the partitions $S$, which are then used to choose the mixture component closest to the target distribution. Assuming Equation 9 is evaluated to determine if this is the case, I am not certain how this would work in practice: how are the J modes found? Is the likelihood evaluated for every datapoint $x$ under each component, to determine the partition it belongs to, before HMC is performed?
> >
> > Section 4.2 is still basically impenetrable. Although there are some lines of explanation, I am still unable to follow the proof for the following reasons:
> > - the notation switches between $V, X$, and $\omega$, and it is only at the end of the section that the reader is told that the 2 are equivalent.
> > - I am still unable to connect the different parts of the proof with the background information or section 4.1, because the proof still mostly skips from equation to equation without providing any intuition for what the goal of each step is.
> > My recommendation would be to sketch out the proof for the reader i.e. "We show ABC by doing DEF. The proof proceeds by showing that GHI is related to JKL" or something along these lines, before the steps of the proof are presented -- added lines simply explaining the equations do not help.
> > - The inline equation at the end of this section is still doing a lot of heavy-lifting. The reader is forced to wade through equations 14-18 which are more or less rearranging terms, but must work out for themselves how the inline equation proves convergence. If this is caused by a lack of space, I would recommend writing a detailed proof in the appendix.
> >
> > The results with folded probGAN are interesting -- thank you for adding these to manuscript (please note: the colours for GAN and folded probGAN are hard to distinguish -- maybe a different color, and continuous lines would be better).
> >
> > Overall however, I will keep my score the same, since my major concerns remain: both sections 4.1 and 4.2, which form the core of the paper, are hard to understand.

---

### Official Review · Reviewer_PuGd · 2021-10-29

**Correctness:** 1
**Technical Novelty And Significance:** 1
**Empirical Novelty And Significance:** 2
**Recommendation:** 1
**Confidence:** 4

**Main Review:**

The paper is not clear, especially for the core part of FHMC formulation.

1. How SGHMC is applied to the S components are not clearly described. I suggest a mathematical formulation instead of using only words.

2. $\mu$ and $\sigma$ are used without any definition beforehand, not sure which parameters they represent.

3. How are these parameters merged for F-HMC?

The theory part is also problematic. The stationary distribution of $V$ (equation 8) should be $\mathcal{N}(0, M)$. $\mathcal{N}(MB^{-1}\nabla U(X), M)$ is the stationary distribution for the second equation in Equation 7. If this is used, the scheme would reduce to SGLD, not SGHMC. Moreover, the proof after that seems to be identical for SGHMC, with a simple dimension decomposition.

The experiments seems to be fine, if done properly. Compared to SGHMC, using F-HMC for Bayes GAN does provide some marginal improvement, not sure if it is statistically significant though. Better provide standard deviations together with the mean estimates.

**Summary Of The Paper:**

This paper proposed Folded Hamiltonian Monte Carlo (FHMC) for posterior sampling over the generator and discriminator parameters in Bayesian GAN.

**Summary Of The Review:**

The main contribution, the FHMC algorithm, is not presented clearly. The same for the theoratical justification. The paper would be strengthened if a formal and more detailed derivation of the proposed methodology is presented.

---

> ### Author Response · Authors · 2021-11-22
> **Response to AnonReviewer2**
>
> We would like to thank the reviewer for their insightful feedback. Thank you for providing explanations and revealing the weak points that help us improve the work. We have clarified the question raised by the reviewer as follow:
>
> 1. paper is not clear, especially the core of F-HMC formulation
>
> We appreciate the reviewer's concern about clarifying the paper's core and explaining the F-HMC in the formula instead of words. To address that, we have updated sections 4.1, 4.2 to present the mathematical formulation of F-HMC, including how we divide the data in S and more details of parameter μ and σ and how we merge them back. We have listed them here for your convenience.
>
> 2. How SGHMC is applied to the S components, what is the μ and σ definition:
>
> F-HMC assumes that the target density $pi$ is approximated using a mixture of Gaussians.  In that case, it is possible to affirm that each mixture component represents a reliable proposal in a given region of the sample space $S⊂R^d$.  To clarify the idea, suppose $pi$ has J modes and consider its approximation by the mixture model as follow:
>
> $q(x) = \sum_{j=1}^{J} \mathcal{N}_d (x;\mu^j, \sigma^j)$                 (eq.1)
>
> where
> $\mathcal{N}_d(x;\mu^j,\sigma^j)$
>
> is the probability density of a d-variate Gaussian distribution with mean μ and covariance matrix σ.  Using the mixture representation (eq.1), we define the partition
>  $\hat{S} = \cup_{k=1}^J S^{j}$   so that $pi$ is more similar to $\mathcal{N}_d (x;\mu^j, \sigma^j)$ on each set  $S^{(j)}$ than any other distribution entering (eq.1) which is formulated as follow:
>
> $S_n^{(j)} = [x: argmax \mathcal{N}(x;\mu_n^{(j')}, \sigma_n^{(j')}) = j]$            (eq.2)
>
>
>
>
> 3. How these parameters are merged for F_HMC
>
> The approximation (eq.1) along with the partition defined in (eq.2) allows us to determine the proposal distribution of F-HMC.
> One may consider proposal distribution as:
>     $Q(x) = \sum_{j=1}^{J} S_n^{(j)} \mathcal{N}(x,\mu_n^{(j)},\sigma_n^{(j)})$
> In other words, we would utilise the dominating component of the mixture as a proposed distribution in each area $S_n^{(j)}$. While such a concept may have acceptable local properties, it may not ensure optimal flow across various regions. Therefore, we are using a second or (fold) HMC on top of these samplers concerning the cross-correlation of partitions to allow a flow between different regions. Thus the F-HMC proposal distribution models are as follow:
>      $Q'(x) = \sum_{j=1}^{J} Q(x) \mathcal{N}(x,\hat{\mu}^{(j)},\hat{\sigma}^{(j)})$
> where $\hat{\sigma}$ is calculated with respect to cross-correlation between all components.
>
>
>
> 4. Equation 8 in the manuscript is the stationary distribution of the second part of equation 7 not the stationary distribution of V and if it is used, this is SGLD, not SGHMC
>
> We appreciate the reviewer's comment, and we have updated the paper to avoid confusion. Equation 8 is the stationary distribution of the second part of equation 7, and equation 7 is SGHMC. that we have derived from the SGHMC paper at ref [1] section 3.2 equation 9.
>
> 5. Proof is identical to SGHMC with simple decomposition
>
> Thank you for pointing out this concern; we have used the similar technique that SGHMC used to prove their model. Our goal is to show that the F-HMC sampler also converges to the same distribution that SGHMC does.
>
> 6. Experiments are fine better to provide standard deviations with the mean estimates
> Thank you for your suggestion. The standard deviation in experiments is given in Table 1.
>
> [1] Chen, Tianqi & Fox, Emily & Guestrin, Carlos. (2014). Stochastic Gradient Hamiltonian

---

### Official Review · Reviewer_Hh37 · 2021-11-03

**Correctness:** 3
**Technical Novelty And Significance:** 3
**Empirical Novelty And Significance:** 3
**Recommendation:** 6
**Confidence:** 3

**Main Review:**


---
Pros:

* The paper proposes a new HMC sampling method for Bayesian-based GAN. I find the method novel and effective.
* The experiments are nice. On the synthetic dataset, the author shows that F-HMC is better than SGHMC when learning Bayesian GAN with high dimensional data. On the natural image dataset, the F-HMC’s effectiveness is further verified.

---
Cons:

* Although the proposed method looks effective, I am not sure about the motivation or insight behind the proposed design. Could the author elaborate on why think of changing SGHMC to F-HMC? What makes F-HMC a better sampler? Does the improvement application-specific? Can F-HMC apply to tasks other than generative modeling?

* Since F-HMC is a general improvement on SGHMC. While ProbGAN improves the learning objective (Posterior formulation) of Bayesian GAN. So I am considering that F-HMC and ProbGAN should be two improvements on the original Bayesian GAN in orthogonal directions. Hence, I suggest the author check the performance of the combination of F-HMC and ProbGAN. Basically, comparing Bayesian GAN + SGHMC v.s. Bayesian GAN + F-HMC (which has been done) and ProbGAN + SGHMC v.s. ProbGAN + F-HMC (which I suggest doing) should give a better picture of the effectiveness of F-HMC.
* The clarity of the paper could be improved. For example, I am not sure how the equation (8) is derived. In the derivation of equation (11), the author mentions Fokker-Planck Equation without explaining it. It would be nice if the author could articulate more on the background knowledge and equation derivation to make the text more accessible to the general audience.

---
Some typos:
* Page 5, equation (13) misses “[”.


**Summary Of The Paper:**

The paper provides a new technique, called Folded HMC, to better sampling for the Bayesian treatments of GANs. The effectiveness of the method is demonstrated on several vision datasets, CIFAR-10, SVHN, ImageNet.

**Summary Of The Review:**

Overall, I vote for accepting. I like the idea of paralleling the HMC sampling for Bayesian-based GAN methods. My major concern is about the clarity of the paper and some additional ablation (see cons above). Hopefully, the authors can address my concern in the rebuttal period.

---

> ### Author Response · Authors · 2021-11-22
> **Response to AnonReviewer1**
>
> Thanks for your positive feedback. We have incorporated many of your suggestions in the updated paper, and please see below for our detailed response and we hope that would resolve any concerns you had:
>
> 1. Motivation and insight behind the design:
>
> Our intention is to utilise different samples for generator parameters to mitigate GAN collapse mode and produce data samples with an adequate degree of entropy, especially when working with high-dimensional and highly correlated and multi-modal data.
>
>
> 2. Why think of changing SGHMC to F-HMC?
>
> Because SGHMC suffers in efficiency when target density is multi-modal and high dimension with high correlation, ref [1] also discusses this issue. To tackle this challenge, we propose a Folded Hamiltonian Monte Carlo (F-HMC) model to replace the SGHMC component of the Bayesian framework.
>
> 3. What makes the F-HMC better sampler?
>
> The F-HMC 's structure makes the F-HMC a preferred sampler, by combining samples from several chains rather than finding a single chain that samples from the whole distribution. Chains are running in parallel, each exploring a different region of the state space (e.g., a few modes only). Our design has the following advantages:
>
>      a.	F-HMC explores more accurately the target density, especially in the scenario with high-dimensional and highly correlated data.
>
>      b.	F-HMC converges faster to the target density in terms of lag number.
>
>      c.	F-HMC provides the practical advantage to the Bayesian GAN method by exploring a rich multi-modal distribution over the weight parameters of generators at an acceptable entropy level.
>
>      d.	More importantly, F-HMC has an efficient run-time in high-dimensional data approximation because of the parallel composition.
>
> 4. Is the design application-specific and can it be applied to tasks other than generative modelling?
>
> The design is not application-specific, and we define it in a general and applicable way. For instance, we can observe that research by ref [2] uses a similar technique on other tasks such as data imputation and enriching data quality.
>
> 5. Checking the performance of the combination of F-HMC and probGAN
>
> Many thanks for your insightful suggestion. We have conducted the experiment by comparing the result with the probGAN + F-HMC with the existing experiment. From Figure 5 in the revised paper we observe that combining the probGAN + F-HMC enhances results. We are running additional experiments and will add the results once they are available.
>
> 6. Clarity in section 4.2 and giving more background on how formulas went from 8 to 11
>
> Thank you for pointing out this problem. We have updated the paper's core in sections 4, 4.1 and 4.2, with more details and background of how formula 8 concluded to 11.
>
> 7. Minor:
> We appreciate the reviewer's comment and feedback on the Minor section. We have addressed them and updated the manuscript based on them.
>
> References:
> [1] Ye, Nanyang and Zhanxing Zhu. "Bayesian Adversarial Learning." NeurIPS (2018).
>
> [2] N. Pourshahrokhi et al data centric AI workshop, NeurIPS (2021)

---

> > ### Comment · Reviewer_Hh37 · 2021-11-30
> > **Thank you**
> >
> > Thank you to the authors for the responses and the updated manuscript.
> >
> > After reading your response and other reviewers' comments, I feel that the clarity of presenting F-HMC still needs to be improved. Due to this reason, I am sorry that I downgrade my score to 5.
> >
> > With that being, I have to acknowledge the author's effort in your response and the revision. Clearly, the newly added sections 4.1, 4.2 in your revision are making it more clear. But it is still not enough. I sincerely suggest the author follow Reviewer jjmZ's detailed feedbacks to further improve the clarity of your work.
> >
> > I also appreciate that the author added the experiments about "probGAN + F-HMC". I am glad to hear that `combining the probGAN + F-HMC enhances results`. Looking forward to seeing your full results on it.
> >
> > Although, I decide to vote for rejecting this work currently. As you may see, from my original comments, I like this work. I believe there is a novelty in your method. The empirical results are promising. I would like to encourage the author to further improve your clarity of presenting method. Good luck with your revision!

---

### Author Response · Authors · 2021-11-22
**General Response**

We thank all the reviewers for the insightful comments and helpful suggestions. Here we summarise the significant changes we made in the revision of our paper. We have updated sections 4.1 and 4.2 to give more details about the design core body and more background and information about the mathematical proof. We have updated Figure 5 with a new result based on suggestion R’1. We have emphasised Figure 1 to avoid confusion about the structure of our model.

---

### Decision · Program_Chairs · 2022-01-20

**Decision:**

Reject

**Comment:**

The paper proposes a method to sample the parameters of the generator and discriminator in a BayesGAN (Saatci and Wilson, 2017) setting. The main innovation is a modified Hamiltonian Monte Carlo sampling scheme. Unfortunately the method is not clearly presented, to the point that all reviewers had difficulties understanding how the method works. The revision is making progress but still does not clearly explain the method. While the paper cannot be accepted for publication in its present form, the experimental results are encouraging so I encourage the authors to keep improving their manuscript.